# Decoupling feature extraction from policy learning: Assessing benefits of state representation learning in goal based robotics

## Abstract

Scaling end-to-end reinforcement learning to control real robots from vision presents a series of challenges, in particular in terms of sample efficiency. Against end-to-end learning, state representation learning can help learn a compact, efficient and relevant representation of states that speeds up policy learning, reducing the number of samples needed, and that is easier to interpret. We evaluate several state representation learning methods on goal based robotics tasks and propose a new unsupervised model that stacks representations and combines strengths of several of these approaches. This method encodes all the relevant features, performs on par or better than end-to-end learning, and is robust to hyper-parameters change.

## 1 Introduction

A common strategy to learn controllers in robotics is to design a reward function that defines the task and search for a policy that maximizes the collected rewards with a Reinforcement Learning (RL) approach.

In RL, the controlled system (environment and robot) is defined by a state $s_t$, i.e., the relevant variables for a controller, often of low dimension (e.g., positions of a robot and a target). At a given state $s_t$, the agent will receive an *observation* $o_t$ from the environment and a reward $r_t$. In some applications, the observation may be directly the state, but in the general case, the observation is raw sensor data (e.g., images from the robot camera). RL must then learn a policy that takes observations as input and returns the action $a_t$ that maximizes expected return.

When the state is not directly accessible, RL should recover it from the observation to learn a good control policy. This could be learned implicitly by an end-to-end approach (cf Fig. 1), i.e. by learning a policy from observation to action, or explicitly by, at first, extracting a representation of this state from the observation and then learning the policy from it.

State representation learning (SRL) (Lesort et al., 2018) aims at learning those states as a compact representation from raw observations and without explicit supervision. One key goal of learning state representation separately from learning the policy is to improve the sample efficiency of the full process by reducing the search space. Indeed, end-to-end approaches, even if adequate for simulation settings, are often not sample efficient enough for real life learning as sampling observations from the environment is particularly costly and time consuming in robotics. Another crucial advantage of reducing the search space is to improve stability of policy learning, a common issue in RL (Henderson et al., 2017).

Although SRL is not restricted to robotics, in this paper, we demonstrate its utility in goal-based robotics tasks, i.e. the controlled agent is a robot, the reward is sparse and only depends on the previous state and taken action, not on a succession of states (therefore excluding tasks like walking or running).

Several approaches exist for SRL that differ in the information they can encode. This paper aims at investigating the benefit of different ways of combining state of the art SRL approaches on policy learning for various goal based robotics tasks. The contributions of this paper are:

- we show the usefulness of decoupling feature extraction from policy learning (Section 5.4) and that random features provide a good baseline

- we propose a new way of combining approaches by stacking state representations instead of mixing them, that allows to mitigate the problem of conflicting objectives and favors disentanglement (Section 4.4)

- we investigate the influence of hyper-parameters when learning a state representation (Section 5.5)

This paper is organized the following way: we first introduce the state of the art in SRL for robotics (Section 2), and clarify how we define an appropriate state representation (Section 3). Then, we explain how we designed our SRL combination approach (Section 4). Finally, we justify and illustrate our approach with experiments in various simulated robotics tasks (Section 5).

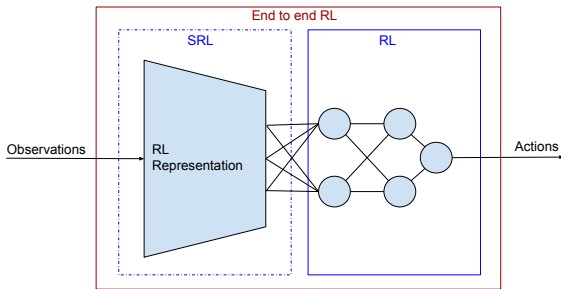

Figure 1: State Representation Learning (SRL) vs End-to-End Reinforcement Learning. In End-to-End learning, the feature extraction is implicit.

## 2 RELATED WORK

In reinforcement learning, a classic preparatory approach is to design some features by hand, in order to facilitate policy learning. However the manual design may be difficult, laborious and requires domain knowledge. Hence, this process can be automated using methods that are able to learn these features (also called representations) (Böhmer et al., 2015; Singh et al., 2012; Boots et al., 2011). This problem is commonly called *State Representation Learning* (SRL). We can define it more precisely as a particular kind of representation learning where the learned features are in low dimension, evolve through time, and are influenced by the actions of an agent (Lesort et al., 2018).

SRL is used as a preliminary step for learning a control policy. The representation is learned on data gathered in the environment by an exploration policy. One particular advantage of this step is that it reduces the search space and gives to reinforcement learning an informative representation, instead of raw data (e.g. pixels). This allows to solve tasks more efficiently (Lange et al., 2012; Wahlström et al., 2015; Munk et al., 2016). We refer to (Lesort et al., 2018) for a complete review of SRL for control.

In robotics, SRL is particularly interesting as the learning process is very slow and data hungry. With real robots, learning happens in real time and cannot be accelerated or easily multiprocessed as in simulated environments. However, since the learning process remains time consuming and the availability of robots is limited by cost and maintenance constraints, several approaches prefer to iterate at first in simulation to learn robotics tasks (Jonschkowski & Brock, 2015; Watter et al., 2015; Curran et al., 2016; Lesort et al., 2017; Jonschkowski et al., 2017). Our proposal is based along this line.

The robotics environment provides us with rewards, observations and actions, that can be used to define SRL loss functions. *Forward models* (Munk et al., 2016), *inverse models* (Shelhamer et al., 2017), *data-reconstruction models* (Mattner et al., 2012; Curran et al., 2016) or *priors knowledge* (Jonschkowski & Brock, 2015; Lesort et al., 2017) are several approaches that exploit those environments data to learn meaningful representations. These methods can also be combined to improve the quality of the learned representations. Some examples include mixing a data-reconstruction objective and a forward model loss (Watter et al., 2015; Krishnan et al., 2015; Ha & Schmidhuber, 2018), coupling a forward model together with an inverse model (Pathak et al., 2017), and using both data-reconstruction and priors loss functions (Finn et al., 2015). The goal of this paper is thus to compare decoupling feature extraction (SRL) from end-to-end policy learning, and to explore various possible combinations to learn these features.

Our setting is similar to the one used in Hindsight Experience Replay (Andrychowicz et al., 2017) that tackles the problem of solving goal-based robotics tasks with sparse reward. In their experiments, the agent has a direct access to the positions of the controlled robot and target. Our work, on the contrary, uses the raw pixels as input. The extraction of relevant positions must be learned by the different methods.

## 3 STATE REPRESENTATION REQUIREMENTS

SRL aims at extracting relevant information from raw sensor data. This ability is not the only substantial characteristic of a SRL model. In this section, we provide additional important facets of a good state space.

From a high-level point of view, the state representation should retain useful information from the observation in order to solve the task and filter out irrelevant parts. More precisely, the state space should be:

*Compact*: a good state representation should have a low dimension compared to the raw sensor data. It should only keep relevant information, ignoring distractors (irrelevant parts of the observation). This will reduce the search space for RL, leading to a more stable and sample-efficient policy learning. A low-dimensional space is also easier to interpret.

*Sufficient*: all the important information to solve the task should be encoded into the state space. Otherwise, the agent will under-perform (cannot reach maximal performance) or even fail.

*Disentangled*: the state representation should untangle factors of variation independently, i.e., each dimension of the feature space should represent a particular factor of variation. A disentangled state representation should facilitate policy learning (because the policy network does not have to learn how to decipher the raw data). The non-redundancy of representation is preferable for disentanglement but not necessary, as long as the representation dimension stays low.

*Generalizable*: As for any machine learning algorithm, representations learned by SRL should be able to generalize to unseen situations. In the SRL context, we can define 2 different degrees of generalization: 1) generalization to new environment states and 2) generalization to different environments.

In the context of a goal-based robotics task, a *sufficient* state representation should extract the position of the robot, and the position of the goal. If velocities are also needed, they can be approximated using finite differences between two consecutive positions, as in (Jonschkowski et al., 2017). A *disentangled* feature space should encode only one coordinate per dimension, i.e., one dimension should encode the x-coordinate of the robot position, another one the y-coordinate, etc.

## 4    INCREMENTALLY BUILDING A POTENTIAL ADEQUATE METHOD

Given the general objectives defined in the previous section, we now propose a way to combine several approaches by tackling one objective at a time, using a particular context for a concrete illustration. This part aims at giving insights on the different SRL methods, taking advantage of goal-based robotics tasks as an application example.

### 4.1    ENCODING STATE OF THE AGENT: ROBOT POSITION

One important aspect to encode for RL is the state of the controlled agent. In the context of goal-based robotics tasks, it corresponds to the robot position. A simple method consists of using an *inverse dynamics objective*: given the current $s_t$ and next state $s_{t+1}$, the task is to predict the taken action $a_t$. The type of dynamics learned is constrained by the network architecture. For instance, using a linear model imposes linear dynamics.

The state representation learned encodes only controllable elements of the environment. Here, the robot is part of them. However, the features extracted by an inverse model are not always *sufficient*: in our case, they do not encode the position of the target since the agent cannot act on it.

### 4.2    ENCODING ADDITIONAL INFORMATION: TARGET POSITION

Since learning to extract the robot position is not enough to solve goal-based tasks, we need to add extra objective functions in order to encode the position of the target object. In this section, we consider two of them: minimizing a reconstruction error (auto-encoder model) or a reward prediction loss.

*Auto-encoder:* Thanks to their reconstruction objective, auto-encoders compress information in their latent space. Auto-encoders tend to encode only aspects of the environment that are salient in the input image. This means they are not task-specific: relevant elements can be ignored and distractors (unnecessary information) can be encoded into the state representation. They usually need more dimensions that apparently required to encode a scene (e.g. in our experiments, it requires more than 10 dimensions to encode a 2D position).

*Reward prediction:* The objective of a reward prediction module leads to state representations that are specialized in a task. However, this does not constrain the feature space to be disentangled or to have any particular structure.

### 4.3 COMBINING APPROACHES

Combining objectives makes it possible to share the strengths of each model. In our application example, the previous sections suggest that we should mix objectives to encode both robot and target positions.

The simplest way to combine objectives is to minimize a weighted sum of the different loss functions, i.e. reconstruction, inverse dynamics and reward prediction losses:

$$\mathcal{L}_{combination} = w_{reconstruction} \cdot \mathcal{L}_{reconstruction} + w_{inverse} \cdot \mathcal{L}_{inverse} + w_{reward} \cdot \mathcal{L}_{reward} \quad (1)$$

Each weight represents the relative importance we give to the different objectives. Because we consider each objective to be relevant, we chose the weights such that they provide gradients with similar magnitudes.

### 4.4 SPLITTING INSTEAD OF COMBINING STATE REPRESENTATIONS

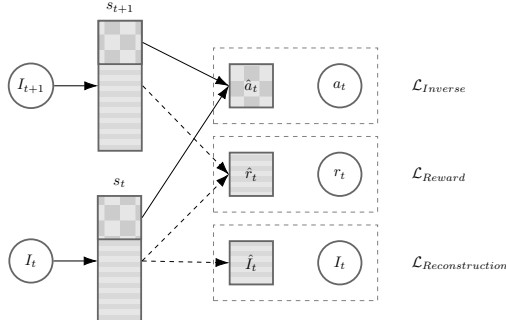

Figure 2: *SRL Splits* model: combines a reconstruction of an image $I$, a reward ($r$) prediction and an inverse dynamic models losses, using two splits of the state representation $s$. Arrows represent model learning and inference, dashed frames represent losses computation, rectangles are state representations, circles are real observed data, and squares are model predictions.

Combining objectives into a single embedding is not the only option to have features that are *sufficient* to solve the tasks. Stacking representations, which also favors *disentanglement*, is another way of solving the problem. We use this idea in the *SRL Splits* model, where the state representation is split into several parts where each optimizes a fraction of the objectives. This prevents objectives that can be opposed from cancelling out and allows a more stable optimization. This process is similar to training several models but with a shared feature extractor, that projects the observations into the state representation.

In practice, as showed in Fig. 2, each loss is only applied to part of the state representation. In the experiments, to encode both target and robot positions, we combine the strength of auto-encoders, reward and inverse losses using a state representation of dimension 200. The reconstruction loss is the mean squared error between original and reconstructed image. We used cross entropy loss for the reward prediction and inverse dynamics losses. The reconstruction and reward losses[1] are applied on a first split of 198 dimensions and the inverse dynamics loss on the 2 remaining dimensions (encoding the robot position). To have the same magnitude for each loss, we set $w_{reconstruction} = 1$, $w_{reward} = 1$ and $w_{inverse} = 2$.

The choice of the different hyper-parameters (losses, weights, state dimension, training-set-size) and the robustness to changes are explored and validated in the experiments section (Section 5) and Appendix B.

## 5 EXPERIMENTS AND RESULTS

### 5.1 ENVIRONMENTS

In order to evaluate the methods, we use 4 environments proposed in *S-RL Toolbox* (Raffin et al., 2018) (Fig. 3). These environments of incremental difficulty are specially designed for evaluating SRL methods in a robotics context. The environments are variations of two main settings: a 2D environment with a mobile robot and a 3D environment with a robotic arm. In all settings, there is a controlled robot and one target that is randomly initialized. In the experiments, the robot is controlled using discrete actions (but the approaches we present are not limited to that domain) and the reward is sparse: +1 when reaching the goal, -1 when hitting an obstacle and 0 everywhere else. The four environments used are: 1D/2D random target with mobile robot and random/moving target with robotic arm.

---

[1]We combine an auto-encoder loss with a reward prediction loss to have task-specific features

| Mobile Navigation | Robotic Arm |
|---|---|

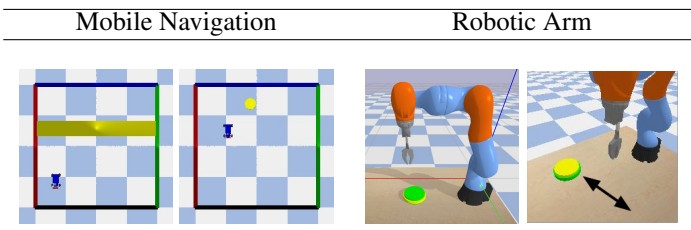

Figure 3: Environments for state representation learning from S-RL toolbox (Raffin et al., 2018)

*1D/2D random target mobile navigation*: This environment consists of a navigation task using a mobile robot, similar to the task in (Jonschkowski & Brock, 2015), with either a cylinder (2D target) or a horizontal band (1D target) on the ground as a goal, randomly initialized at the beginning of each episode. The mobile robot can move in four directions (*forward, backward, left, right*) and will get a +1 reward when reaching the target, -1 when hitting walls, and 0 otherwise. Episodes have a maximum length of 250 steps (hence, an upper bound max. reward of 250).

*Robotic arm with random/moving target*: In this setting, a robotic arm, fixed to a table, has to reach a randomly initialized target on the table. The target can be static during the episode or slowly moving back and forth along one axis. The arm is controlled in the $x$, $y$ and $z$ position using inverse kinematics. The agent received a +1 reward when it reaches the goal, -1 when hitting the table, and 0 otherwise. The episode terminates either when the robot hits the table or when it touches 5 times the target (hence, the max. reward value is 5). Episodes have a maximum length of 1000/1500 steps in the random/moving target settings, respectively.

All environments correspond to a fully observable Markov Decision Process (MDP), i.e., target object and agent are always visible and the next observation $o_{t+1}$ only depends on the previous couple $(o_t, s_t)$ (except for the robot arm setting with moving target where there is small uncertainty for the position of the target).

We chose those specific tasks because they were i) designed for robotics, ii) appropriate for evaluating SRL models iii) of gradual difficulty. Those environments cover basic goal-based robotics tasks: mobile navigation and reaching a 3D position. Then, they are designed with simplicity in mind, making the extracted features easier to interpret. It is also clear what a good state representation should encode because of the small number of relevant elements.

## 5.2 EVALUATION METRICS

We use two methods to evaluate a learned state representation. First, since the main goal of extracting relevant features is to solve a task, we compare performance in Reinforcement Learning. To have quantitative results, each RL experiment uses 10 different random seeds[2]. We chose two metrics: mean reward over 100 episodes at the end of training and mean reward over 100 episodes for a given budget (a fixed number of timesteps). This last metric is particularly relevant when doing robotics: the budget is much more limited than in simulation and we want to reach an acceptable performance as soon as possible.

Then, since we have access to the true positions, we can also compute the correlation between ground truth states and learned states. However, looking at a correlation matrix when the state dimension is large is impractical. Therefore, we use the metric *GroundTruthCorrelation* (*GTC*) described in (Raffin et al., 2018). It measures the maximum correlation (in absolute value) in the learned representation for each dimension of the ground truth (GT) states. *GTC* is defined as:

$$GTC_{(i)} = \max_j |\rho_{s,\tilde{s}}(i,j)| \in [0,1] \tag{2}$$

where $\rho_{s,\tilde{s}}$ is the Pearson correlation coefficient for the pair $(s, \tilde{s})$, where $\tilde{s}$ is the GT state, $s$ the learned state, $i \in [\![0, |\tilde{s}|]\!]$, $j \in [\![0, |s|]\!]$, $\tilde{s} = [\tilde{s}_1; ...; \tilde{s}_n]$, and $\tilde{s}_k$ being the $k^{th}$ dimension of the GT state vector.

*GTC* produces a vector of a fixed size for a given environment making it able to compare different approaches and different representation dimensions.

This measure allows to have some understanding of what was learned. More precisely, *GTC* can provide us with information on the sufficiency: it gives insights of which component of the Ground Truth state was encoded in the state representation. If in addition, the representation is disentangled, then each factor of variation will be encoded by a different component, leading to a *GTC* close to 1.

---

[2]Except the ablation study that uses 5 random seeds

However, this measure is only a proxy to minimally guarantee that the required information is encoded: having a good *GTC* is sufficient to succeed in RL but not necessary. For instance, as described more extensively in Appendix C, the correlation is not invariant to transformations, such as rotation of the frame, even if the variation remains limited. Finally, this metric is not an estimate of the representation compactness since it is independent of the representation dimension.

### 5.3    IMPLEMENTED APPROACH AND BASELINES

We evaluate the two proposed combination methods:

- **SRL Combination** The combination of reconstruction, reward and inverse losses is done by averaging them on a single embedding (Sec. 4.3).
- **SRL Splits** The model described in Sec. 4.4 and Fig. 2 that combines reconstruction, reward and inverse losses using splits of the state representation.

and compare them with several baselines:

- **Raw Pixels** Learning a policy in an end-to-end manner, directly from pixels to actions.
- **Ground Truth (GT)** Hand engineered features: true robot and target object positions.
- **Supervised** A model trained with Ground Truth states as targets in a supervised setting.
- **Random Features** The feature extractor, a convolutional network, is fixed after random initialization.
- **Auto-encoder** We took the best model between auto-encoder (cf. Sec. 4), denoising auto-encoder and Variational Auto-Encoder (VAE) (Kingma & Welling, 2013), which was in our case the *vanilla* one.
- **Robotic Priors** The method described in (Jonschkowski et al., 2017) that encodes prior knowledge about the world as losses[3].

Each state representation has a dimension of 200 and is learned using 20 000 samples collected with a random policy. The implementation and additional training details can be found in Appendix A.

### 5.4    END-TO-END VERSUS STATE REPRESENTATION LEARNING

| Environments | Nav. 1D Target | Nav. 2D Target | Arm Random Target | Arm Moving Target |
|---|---|---|---|---|
| Ground Truth | $211.6 \pm 14.0$ | $234.4 \pm 1.3$ | $4.2 \pm 0.5$ | $4.6 \pm 0.2$ |
| Supervised | $189.7 \pm 14.8$ | $213.5 \pm 6.0$ | $3.1 \pm 0.3$ | $1.4 \pm 0.4$ |
| Raw Pixels | $215.7 \pm 9.6$ | $231.5 \pm 3.1$ | $2.6 \pm 0.3$ | $2.0 \pm 0.3$ |
| Random Features | $211.9 \pm 10.0$ | $208 \pm 6.1$ | $4.1 \pm 0.3$ | $3.0 \pm 0.3$ |
| Auto-Encoder | $188.8 \pm 13.5$ | $192.6 \pm 8.9$ | $3.4 \pm 0.3$ | $3.0 \pm 0.4$ |
| SRL Combination | $216.3 \pm 10.0$ | $183.6 \pm 9.6$ | $2.9 \pm 0.3$ | $2.9 \pm 0.4$ |
| SRL Splits | $205.1 \pm 11.7$ | $232.1 \pm 2.2$ | $3.7 \pm 0.3$ | $2.5 \pm 0.3$ |

Table 1: Mean reward performance and standard error in RL (using PPO) per episode (average on 100 episodes) at the end of training for all the environments tested.

Table 1 displays the mean reward, averaged on 100 episodes, for each environment after RL training. Table 3 shows the RL performance for different budgets on the Robotic Arm with random target task. To compare SRL methods, $GTC$, $GTC_{mean}$ and associated RL performance are displayed in Table 2 for the navigation task with a 2D target. Complete results for all the environments can be found in Appendix B.

For every environment, there is always a SRL method that reaches or exceeds the performance obtained using only the raw pixels as input. SRL methods do not necessary improve final performance (see Table 1), however as shown in Table 3, it is useful to improve the learning speed of the policy. For instance, in the robotic arm task, much more samples are needed to attain similar levels of performance to those achieved by learning in an end-to-end manner.

*SRL Splits* is the approach that performs on par or better than learning from raw pixels across all the tasks. Its counterpart, *SRL Combination*, that uses only a single embedding, gives also positive results, except for the navigation environment with a 2D random target where it under-performs. The *GTC* provides us with

---

[3]Because of the poor results obtained with this method in preliminary experiments, we do not show results on all the environments.

| Ground Truth Correlation | $x_{robot}$ | $y_{robot}$ | $x_{target}$ | $y_{target}$ | Mean | Mean Reward |
|---|---|---|---|---|---|---|
| Ground Truth | 1 | 1 | 1 | 1 | 1 | $234.4 \pm 1.3$ |
| Supervised | 0.69 | 0.73 | 0.70 | 0.72 | 0.71 | $213.5 \pm 6.0$ |
| Random Features | 0.68 | 0.65 | 0.34 | 0.31 | 0.50 | $208 \pm 6.1$ |
| Robotic Priors | 0.2 | 0.2 | 0.41 | 0.66 | 0.37 | $6.2 \pm 3.1$ |
| Auto-Encoder | 0.52 | 0.51 | 0.24 | 0.23 | 0.38 | $192.6 \pm 8.9$ |
| SRL Combination | 0.92 | 0.92 | 0.33 | 0.42 | 0.65 | $183.6 \pm 9.6$ |
| SRL Splits | 0.81 | 0.84 | 0.64 | 0.39 | 0.67 | $232.1 \pm 2.2$ |

Table 2: $GTC$, $GTC_{mean}$, and mean reward performance in RL (using PPO) per episode after 5 millions steps, with standard error (SE) for each SRL method in mobile robot navigation 2D random target environment.

some insights (see Table 2): both methods extract the robot position (absolute correlation close to 1), yet the target position is better encoded with the SRL splits method, which may explain the gap in performance.
In the robotic arm setting, combining approaches does not seem to be of much benefit. Two possible reasons may explains that. First, compared to the mobile robot, the robotic arm and the target are visually salient so an auto-encoder is sufficient to solve the task. Second, the actions magnitude is much smaller in the robotic arm environments, therefore learning an inverse model is much harder in this setting.

| Budget (in timesteps) | 1 Million | 2 Million | 3 Million | 5 Million |
|---|---|---|---|---|
| Ground Truth | $4.1 \pm 0.5$ | $4.1 \pm 0.6$ | $4.1 \pm 0.6$ | $4.2 \pm 0.5$ |
| Supervised | $4.0 \pm 0.3$ | $3.8 \pm 0.3$ | $3.4 \pm 0.3$ | $3.1 \pm 0.3$ |
| Raw Pixels | $0.6 \pm 0.3$ | $0.8 \pm 0.3$ | $1.2 \pm 0.3$ | $2.6 \pm 0.3$ |
| Random Features | $1.5 \pm 0.4$ | $2.8 \pm 0.3$ | $3.5 \pm 0.3$ | $4.1 \pm 0.3$ |
| Auto-Encoder | $0.92 \pm 0.3$ | $1.6 \pm 0.3$ | $2.2 \pm 0.3$ | $3.4 \pm 0.3$ |
| SRL Combination | $1.0 \pm 0.3$ | $1.5 \pm 0.3$ | $2.0 \pm 0.3$ | $2.9 \pm 0.3$ |
| SRL Splits | $1.1 \pm 0.3$ | $2.1 \pm 0.3$ | $2.7 \pm 0.4$ | $3.7 \pm 0.3$ |

Table 3: Mean reward performance in RL (using PPO) per episode (average on 100 episodes) for different budgets, with standard error in robotic arm with random target environment.

*Ground Truth* states naturally outperform all the methods across all environments. This highlights the importance of having a low dimensional and informative representation. The *Supervised* baseline allows to quickly attain an acceptable performance, but then reaches a plateau (e.g. Fig. 6). Compared to the unsupervised methods, it apparently generalizes less efficiently to data not present in the training set.

As in Burda et al. (2018), the *Random Features* model performs decently on all the environments and sometimes better

(cf. Table 3) than learned features. Looking at the *GTC* (Tables 2, 5, 8, 9), random features keep the useful information to solve the tasks. Hence, we hypothesize that random features should work in environments where visual observations are simple enough because they can preserve enough information.

Despite good results in mobile robot navigation with a static target (Jonschkowski & Brock, 2015), *Robotic Priors* are not well suited when the target changes from episode to episode. As described in Lesort et al. (2017), robotics priors lead to a state representation that contains one cluster per episode, which prevent generalization and good performances in these RL tasks.

The *auto-encoder* has mixed results: it allows to solve all environments, yet it under-performs in the navigation tasks. When we explored the latent space using the S-RL Toolbox (Raffin et al., 2018), we noticed that one dimension of the state space could act on both robot and target positions in the reconstructed image. Our hypothesis, also supported by the *GTC*, is that the state space is not disentangled. This approach does not make use of additional information that the environment provides, such as actions and rewards, leading to a latent space that may lack of informative structure.

| $w_{reconstruction}$ | $w_{reward}$ | $w_{inverse}$ | Mean Reward |
|---|---|---|---|
| 1 | 1 | 1 | $225.2 \pm 6.3$ |
| 1 | 1 | 10 | $223.5 \pm 8.0$ |
| 1 | 10 | 10 | $215.1 \pm 7.1$ |
| 1 | 10 | 5 | $217.8 \pm 12.2$ |
| 1 | 5 | 1 | $217.8 \pm 6.7$ |
| 1 | 5 | 10 | $228.8 \pm 4.2$ |
| 5 | 1 | 1 | $221.0 \pm 7.4$ |
| 5 | 1 | 10 | $209.1 \pm 19.5$ |
| 5 | 10 | 10 | $226.3 \pm 5.2$ |
| 5 | 5 | 1 | $194.6 \pm 14.6$ |
| 5 | 5 | 10 | $224.5 \pm 5.5$ |
| 10 | 1 | 1 | $176.5 \pm 16.2$ |
| 10 | 1 | 10 | $218.9 \pm 8.0$ |
| 10 | 1 | 5 | $182.4 \pm 15.8$ |
| 10 | 5 | 10 | $225.5 \pm 5.7$ |
| 10 | 5 | 5 | $210.2 \pm 8.3$ |

Table 4: Influence of the weights for the SRL Splits model performance in Navigation 2D random target environment.

## 5.5 ABLATION
STUDY AND HYPERPARAMETERS INFLUENCE STUDY

To better understand the influence of each hyper-parameter and study the robustness of SRL, we performed a thorough analysis of *SRL Splits* in the mobile robot navigation with 2D random target setting.

Figure 4 (and Table 11 in the appendix) show the result of the ablation study performed on the *SRL Splits* model. As expected,

the inverse model allows to extract the position of the controllable object, which is the robot. This helps to solve the task and results in a performance boost. In the same vein, the addition of a reward loss favors the encoding of the target position. It also does not seem necessary to separate the reconstruction and reward losses as they encode the same information.

Table 4 displays the influence of the weights of the loss combination on the final mean reward. It shows that the method works on a wide range of different weighting schemes, as long as the reconstruction and the inverse loss have similar magnitude. When the reconstruction weight is one order of magnitude greater, the model behaves like an auto-encoder (because the feature extractor is shared).

In the appendix, extra results (Figs. 10, 11 and 12) exhibit the stability and robustness of SRL against additional hyper-parameter changes (random seed, training set size and dimensionality of the state learned). The state dimension needs to be large enough (at least 50 dimensions for the mobile navigation environment), but increasing it further has no incidence on the performance in RL. In a similar way, a minimal number of training samples (10000) is required to efficiently solve the task. Over that limit, adding more samples does not affect the final mean reward.

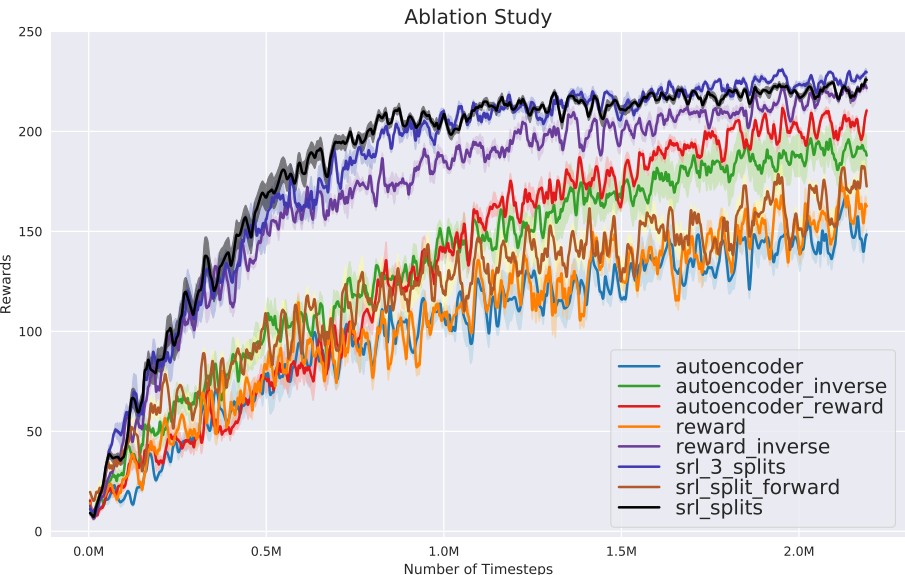

Figure 4: Ablation study of *SRL Splits* (mean and standard error for 10 runs) for PPO algorithm in Navigation 2D random target environment. Models details are explained in Table 11, e.g., *SRL_3_splits* model allocates separate parts of the state representation to each loss (reconstruction/reward/inverse).

During our experiments, we found that learning the policy end-to-end was more sensitive to hyper-parameter changes. For instance, hyper-parameters tuning of A2C (Fig.7) was needed in order to have decent results for the pixels, whereas the performance was stable for the SRL methods. This can be explained by the reduced search space: the task is simpler to solve when features are already extracted. A more in-depth study would be interesting in the future.

## 6 CONCLUSIONS

In this work, we presented the advantages of decoupling feature extraction from policy learning in RL, on a set of goal-based robotics tasks. This decomposition reduces the search space, accelerates training, does not degrade final performances and gives more easily interpretable representations with respect to the true state of the system. We show also that random features provide a good baseline versus end-to-end learning.

We introduced a new way of effectively combining approaches by splitting the state representation. This method uses the strengths of different SRL models and reduces interference between opposed or conflicting objectives when learning a feature extractor. Finally, we showed the influence of hyper-parameters on SRL Split model and the relative robustness of this model against perturbations.

Future work should take advantage of the study done in simulation to experiment those methods on real robots.

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

## A  IMPLEMENTATION DETAILS

Each state representation is learned using 20 000 samples collected using a random policy. We kept for each method the model with the lowest validation loss during the 30 training epochs. We used the same network architecture from (Raffin et al., 2018) for all the models. The input observations of all models are RGB images of size $224 \times 224 \times 3$. Navigation environments use 4 discrete actions (*right, left, forward, backward*); robotic arm environments use one more (*down*) action.

We used PPO (the GPU version called PPO2) and A2C implementations from stable-baselines (Hill et al., 2018), a fork of OpenAI Baselines (Dhariwal et al., 2017). PPO was the RL algorithm that worked well across environments and methods without any hyper-parameter tuning, and therefore, the selected one for our experiments.

Regarding the network learning the policies, the same architecture is used in all different methods. For the approaches that do not use pixels, it is a 2-layers MLP, whereas for learning from raw pixels, it is the CNN from (Mnih et al., 2015) present in OpenAI baselines.

Observations are normalized, either by dividing the input by 255 (for raw pixels) or by computing a running mean/std average (for SRL models).

For the *SRL Splits* and *SRL Combination* methods, we used a linear model for the inverse dynamics, and a 2-layers MLP of 16 units each with ReLU activation for the reward prediction. Only one minor adjustment was made on the *SRL Splits* method for the robotic arm environments: because the controlled robot is more complex, the inverse model was allocated more dimensions (10 instead of 2) but keeping the total state dimension constant (equal to 200).

Code and data to reproduce our results will be released after the reviewing process.

## B  ADDITIONAL RESULTS

### B.1  MOBILE NAVIGATION WITH 1D TARGET

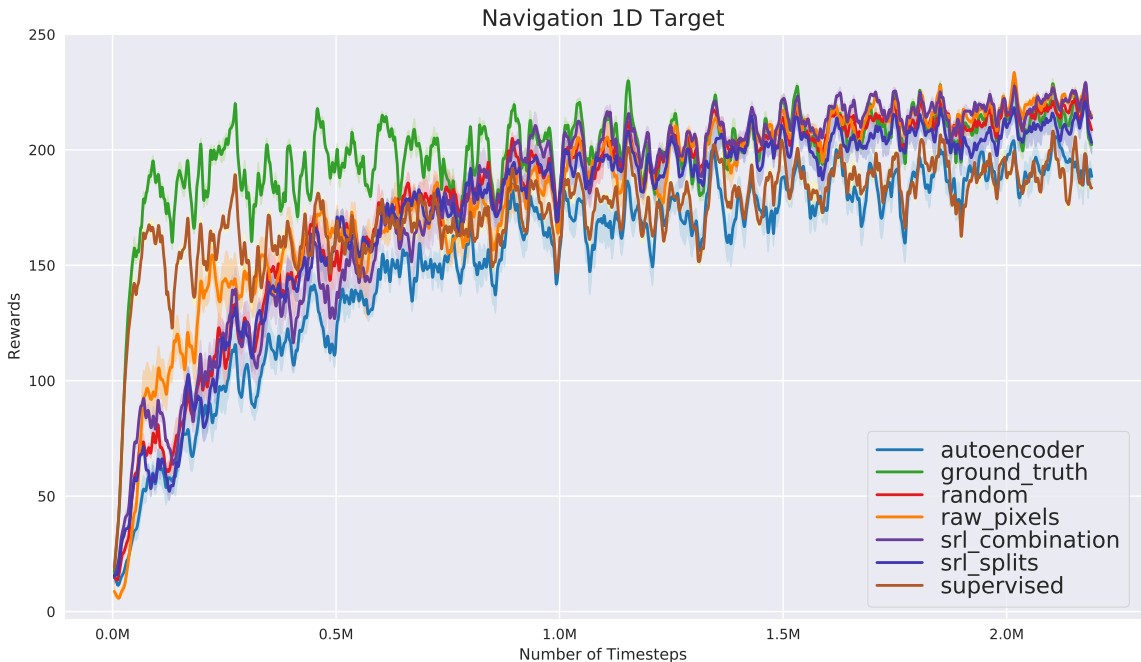

Figure 5: Performance (mean and standard error for 10 runs) for PPO algorithm for different state representations learned in Navigation 1D target environment.

| Ground Truth Correlation | $x_{robot}$ | $y_{robot}$ | $x_{target}$ | Mean | Mean Reward |
|---|---|---|---|---|---|
| Ground Truth | 1 | 1 | 1 | 1 | $211.6 \pm 14.0$ |
| Supervised | 0.68 | 0.77 | 0.72 | 0.72 | $189.7 \pm 14.8$ |
| Random Features | 0.56 | 0.63 | 0.70 | 0.63 | $211.9 \pm 10.0$ |
| Auto-Encoder | 0.36 | 0.46 | 0.77 | 0.53 | $188.8 \pm 13.5$ |
| SRL Combination | 0.95 | 0.98 | 0.39 | 0.77 | $216.3 \pm 10.0$ |
| SRL Splits | 0.81 | 0.92 | 0.79 | 0.84 | $205.1 \pm 11.7$ |

Table 5: $GTC$, $GTC_{mean}$, and mean reward performance in RL (using PPO) per episode after 2 millions steps, with standard error (SE) for each SRL method in Navigation 1D target environment.

| Budget (in timesteps) | 1 Million | 2 Million |
|---|---|---|
| Ground Truth | $198.0 \pm 16.1$ | $211.6 \pm 14.0$ |
| Supervised | $169.5 \pm 13.5$ | $189.7 \pm 14.8$ |
| Raw Pixels | $177.9 \pm 15.6$ | $215.7 \pm 9.6$ |
| Random Features | $187.8 \pm 12.6$ | $211.6 \pm 10.0$ |
| Auto-Encoder | $159.8 \pm 16.1$ | $188.8 \pm 13.5$ |
| SRL Combination | $191.0 \pm 14.2$ | $216.3 \pm 10.0$ |
| SRL Splits | $184.5 \pm 12.3$ | $205.1 \pm 11.7$ |

Table 6: Mean reward performance in RL (using PPO) per episode (average on 100 episodes) for different budgets, with standard error in Navigation 1D target environment.

## B.2 MOBILE NAVIGATION WITH 2D TARGET

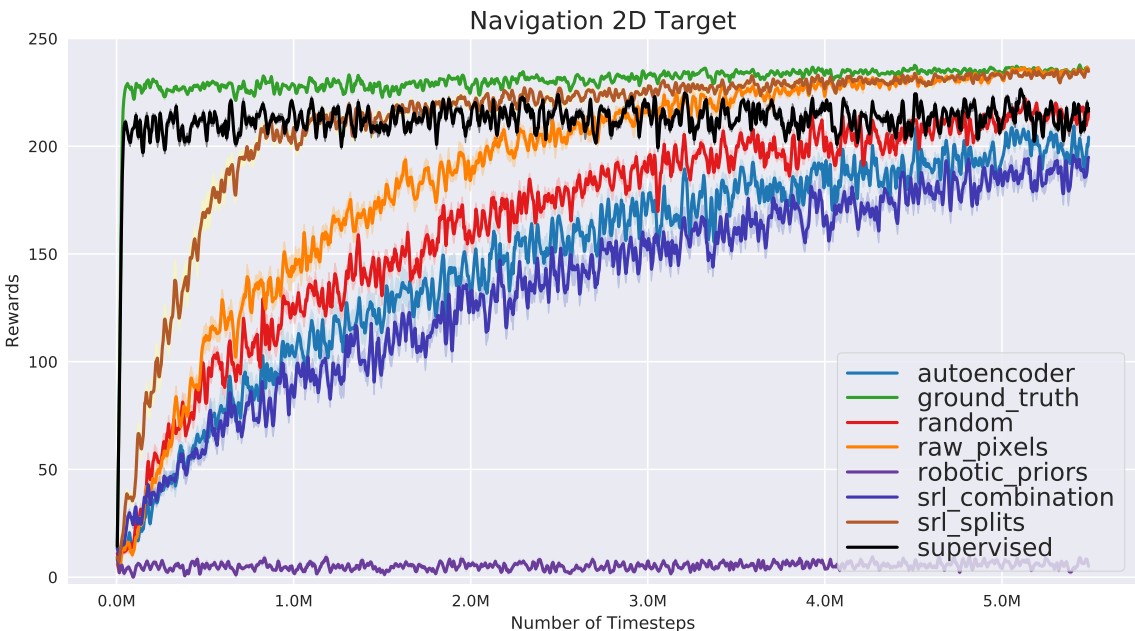

Figure 6: Performance (mean and standard error for 10 runs) for PPO algorithm for different state representations learned in Navigation 2D random target environment.

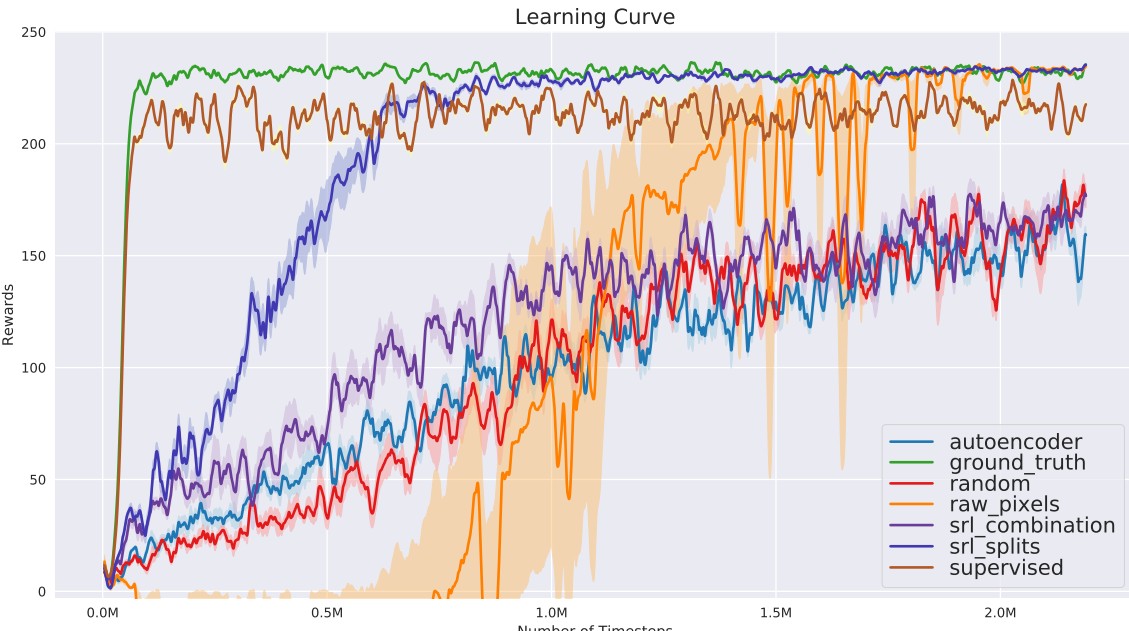

Figure 7: Performance (mean and standard error for 10 runs) for A2C algorithm for different state representations learned in Navigation 2D random target environment.

| **Budget** (in timesteps) | *1 Million* | *2 Million* | *3 Million* | *5 Million* |
|---|---|---|---|---|
| Ground Truth | $227.8 \pm 2.8$ | $229.7 \pm 2.7$ | $231.5 \pm 1.9$ | $234.4 \pm 1.3$ |
| Supervised | $213.1 \pm 6.1$ | $213.3 \pm 6.0$ | $214.7 \pm 5.6$ | $213.5 \pm 6.0$ |
| Raw Pixels | $136.3 \pm 11.5$ | $188.2 \pm 9.4$ | $214.0 \pm 5.9$ | $231.5 \pm 3.1$ |
| Random Features | $116.3 \pm 11.2$ | $163.4 \pm 10.0$ | $186.8 \pm 8.2$ | $208 \pm 6.1$ |
| Robotic Priors | $4.9 \pm 2.9$ | $5.4 \pm 3.1$ | $4.9 \pm 2.8$ | $6.2 \pm 3.1$ |
| Auto-Encoder | $97.0 \pm 12.3$ | $138.5 \pm 12.3$ | $167.7 \pm 11.1$ | $192.6 \pm 8.9$ |
| SRL Combination | $83.9 \pm 40.7$ | $123.1 \pm 11.6$ | $150.1 \pm 11.0$ | $183.6 \pm 9.6$ |
| SRL Splits | $205.5 \pm 6.6$ | $219.5 \pm 5.1$ | $223.4 \pm 4.5$ | $232.1 \pm 2.2$ |

Table 7: Mean reward performance in RL (using PPO) per episode (average on 100 episodes) for different budgets, with standard error in Navigation 2D random target environment.

### B.3 ROBOTIC ARM WITH RANDOMLY INITIALIZED TARGET

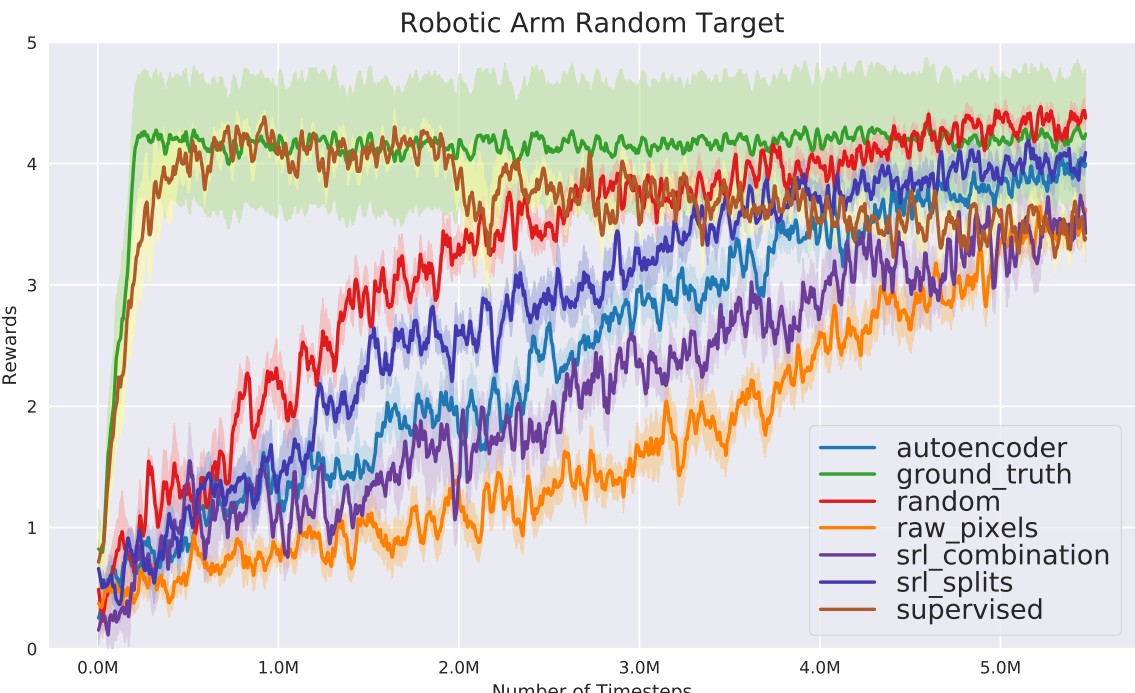

Figure 8: Performance (mean and standard error for 10 runs) for PPO algorithm for different state representations learned in robotic arm with random target environment.

| **Ground Truth Correlation** | $x_{robot}$ | $y_{robot}$ | $z_{robot}$ | $x_{target}$ | $y_{target}$ | *Mean* | *Mean Reward* |
|---|---|---|---|---|---|---|---|
| Ground Truth | 1 | 1 | 1 | 1 | 1 | 1 | $4.2 \pm 0.5$ |
| Supervised | 0.46 | 0.58 | 1.0 | 0.94 | 0.84 | 0.65 | $3.1 \pm 0.3$ |
| Random Features | 0.34 | 0.58 | 0.62 | 0.71 | 0.83 | 0.55 | $4.1 \pm 0.3$ |
| Robotic Priors | 0.21 | 0.18 | 0.42 | 0.7 | 0.66 | 0.38 | $2.6 \pm 0.3$ |
| Auto-Encoder | 0.45 | 0.8 | 0.84 | 0.40 | 0.45 | 0.53 | $3.4 \pm 0.3$ |
| SRL Combination | 0.5 | 0.8 | 0.71 | 0.59 | 0.55 | 0.56 | $2.9 \pm 0.3$ |
| SRL Splits | 0.42 | 0.81 | 0.73 | 0.51 | 0.58 | 0.55 | $3.7 \pm 0.3$ |

Table 8: $GTC$, $GTC_{mean}$, and mean reward performance in RL (using PPO) per episode after 5 millions steps, with standard error for each SRL method in robotic arm with random target environment.

### B.4 ROBOTIC ARM WITH MOVING TARGET

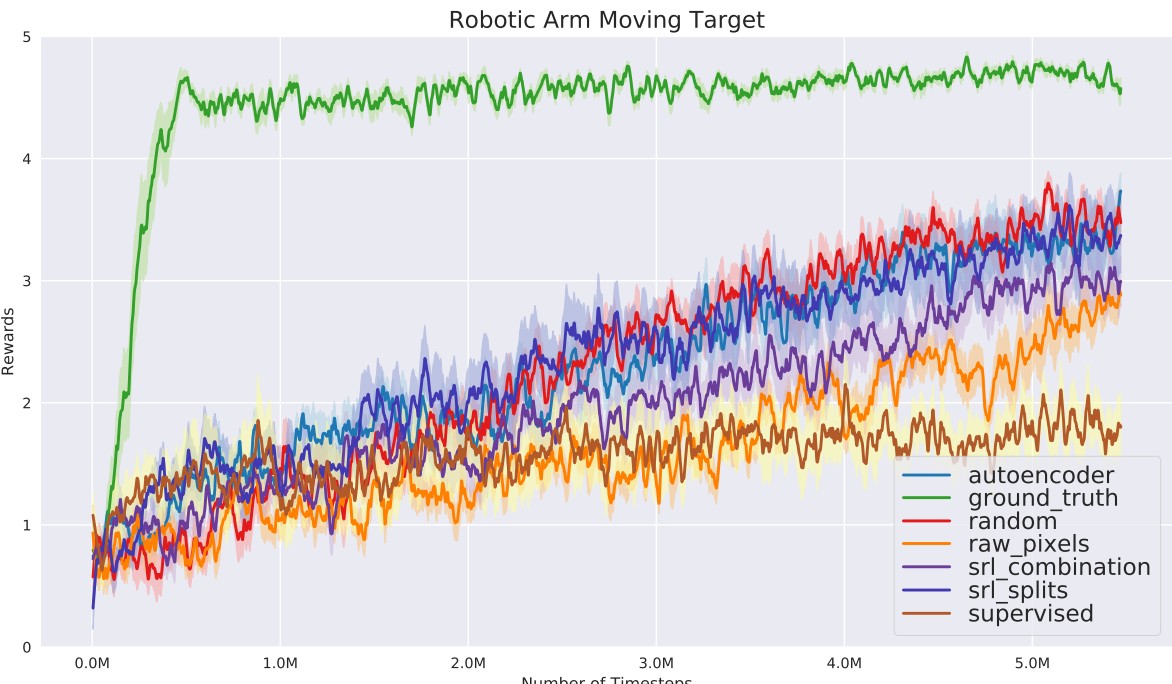

Figure 9: Learning curve (mean and standard error for 10 runs) for PPO algorithm for different state representations learned in robotic arm with random moving target environment.

| **Ground Truth Correlation** | $x_{robot}$ | $y_{robot}$ | $z_{robot}$ | $x_{target}$ | $y_{target}$ | *Mean* | *Mean Reward* |
|---|---|---|---|---|---|---|---|
| Ground Truth | 1 | 1 | 1 | 1 | 1 | 1 | $4.6 \pm 0.2$ |
| Supervised | 0.41 | 0.66 | 1.0 | 0.94 | 0.90 | 0.69 | $3.1 \pm 0.3$ |
| Random Features | 0.35 | 0.59 | 0.66 | 0.73 | 0.3 | 0.49 | $3.0 \pm 0.3$ |
| Auto-Encoder | 0.40 | 0.81 | 0.80 | 0.66 | 0.27 | 0.53 | $3.0 \pm 0.4$ |
| SRL Combination | 0.47 | 0.72 | 0.72 | 0.61 | 0.26 | 0.50 | $2.9 \pm 0.4$ |
| SRL Splits | 0.4 | 0.75 | 0.76 | 0.56 | 0.26 | 0.50 | $2.5 \pm 0.4$ |

Table 9: $GTC$, $GTC_{mean}$, and mean reward performance in RL (using PPO) per episode after 5M steps, with standard error (SE) for each SRL method in robotic arm with moving target environment.

| **Budget** (in timesteps) | *1 Million* | *2 Million* | *3 Million* | *5 Million* |
|---|---|---|---|---|
| Ground Truth | $4.3 \pm 0.3$ | $4.4 \pm 0.2$ | $4.4 \pm 0.2$ | $4.6 \pm 0.2$ |
| Supervised | $1.2 \pm 0.4$ | $1.3 \pm 0.4$ | $1.3 \pm 0.4$ | $1.4 \pm 0.4$ |
| Raw Pixels | $0.8 \pm 0.3$ | $1.0 \pm 0.3$ | $1.2 \pm 0.3$ | $2.0 \pm 0.3$ |
| Random Features | $0.9 \pm 0.3$ | $1.4 \pm 0.3$ | $2.1 \pm 0.3$ | $3.0 \pm 0.3$ |
| Auto-Encoder | $1.17 \pm 0.3$ | $1.5 \pm 0.3$ | $1.9 \pm 0.4$ | $3.0 \pm 0.4$ |
| SRL Combination | $1.2 \pm 0.3$ | $1.6 \pm 0.3$ | $2.2 \pm 0.4$ | $2.9 \pm 0.4$ |
| SRL Splits | $1.0 \pm 0.3$ | $1.3 \pm 0.3$ | $1.7 \pm 0.3$ | $2.5 \pm 0.4$ |

Table 10: Mean reward performance in RL (using PPO) per episode (average on 100 episodes) for different budgets, with standard error in robotic arm with moving target environment.

## B.5 ABLATION STUDY

| Ground Truth Correlation | $x_{robot}$ | $y_{robot}$ | $x_{target}$ | $y_{target}$ | Mean | Mean reward |
|---|---|---|---|---|---|---|
| Auto-Encoder | 0.52 | 0.51 | 0.24 | 0.23 | 0.38 | $138.5 \pm 12.3$ |
| Auto-Encoder / Inverse | 0.94 | 0.94 | 0.37 | 0.40 | 0.66 | $185.4 \pm 16.4$ |
| Auto-Encoder + Reward | 0.41 | 0.37 | 0.70 | 0.46 | 0.48 | $200.7 \pm 10.1$ |
| Reward | 0.57 | 0.43 | 0.32 | 0.57 | 0.47 | $150.1 \pm 15.2$ |
| Reward / Inverse | 0.85 | 0.92 | 0.48 | 0.67 | 0.73 | $211 \pm 8.2$ |
| Auto-Encoder / Reward / Inverse (SRL 3 Splits) | 0.92 | 0.89 | 0.51 | 0.59 | 0.73 | $223.4 \pm 5.6$ |
| Auto-Encoder + Reward / Inverse (SRL Splits) | 0.81 | 0.84 | 0.64 | 0.39 | 0.67 | $232.1 \pm 2.2$ |
| Auto-Encoder + Reward / Inverse + Forward | 0.99 | 0.99 | 0.31 | 0.33 | 0.66 | $159.6 \pm 15.1$ |
| Raw Pixels | N/A | N/A | N/A | N/A | N/A | $188.2 \pm 9.5$ |

Table 11: $GTC$, $GTC_{mean}$, and mean reward performance in RL (using PPO) per episode after 2 millions steps, with standard error for each SRL method in Navigation 2D random target environment. The slash / stands for using different splits of the state representation, and the plus + for combining methods on a shared representation; e.g *Auto-Encoder + Reward* stands for combining an Auto-Encoder to a Reward model. whereas *Auto-Encoder / Reward* means that each loss applies on a separate part of the state representation.

## B.6 INFLUENCE OF THE RANDOM SEED

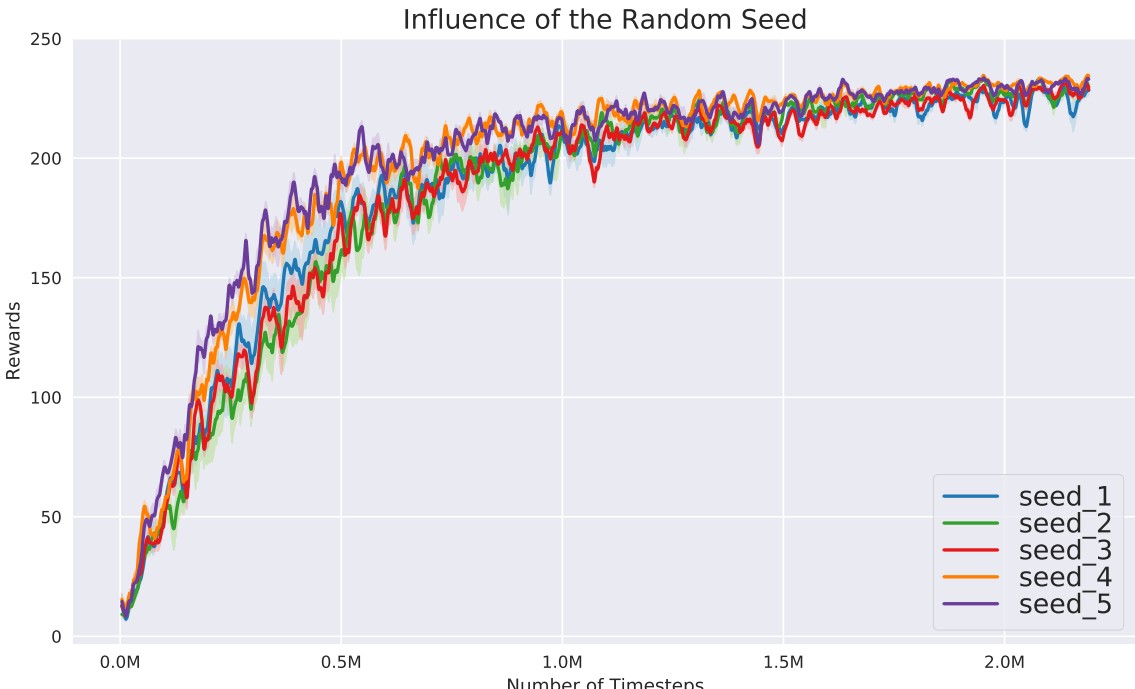

Figure 10: Influence of random seed (mean and standard error for 10 runs) for PPO algorithm for SRL Splits in Navigation 2D random target environment

Figure 10 shows that the SRL *Split* method is stable and its performance does not depend on the random seed used.

## B.7 INFLUENCE OF THE STATE DIMENSION

As shown in figure 11, the state dimension for the SRL model needs to be large enough in order to efficiently solve the task. However, over a threshold, increasing the state dimension does not affect (positively or negatively) the performance in RL.

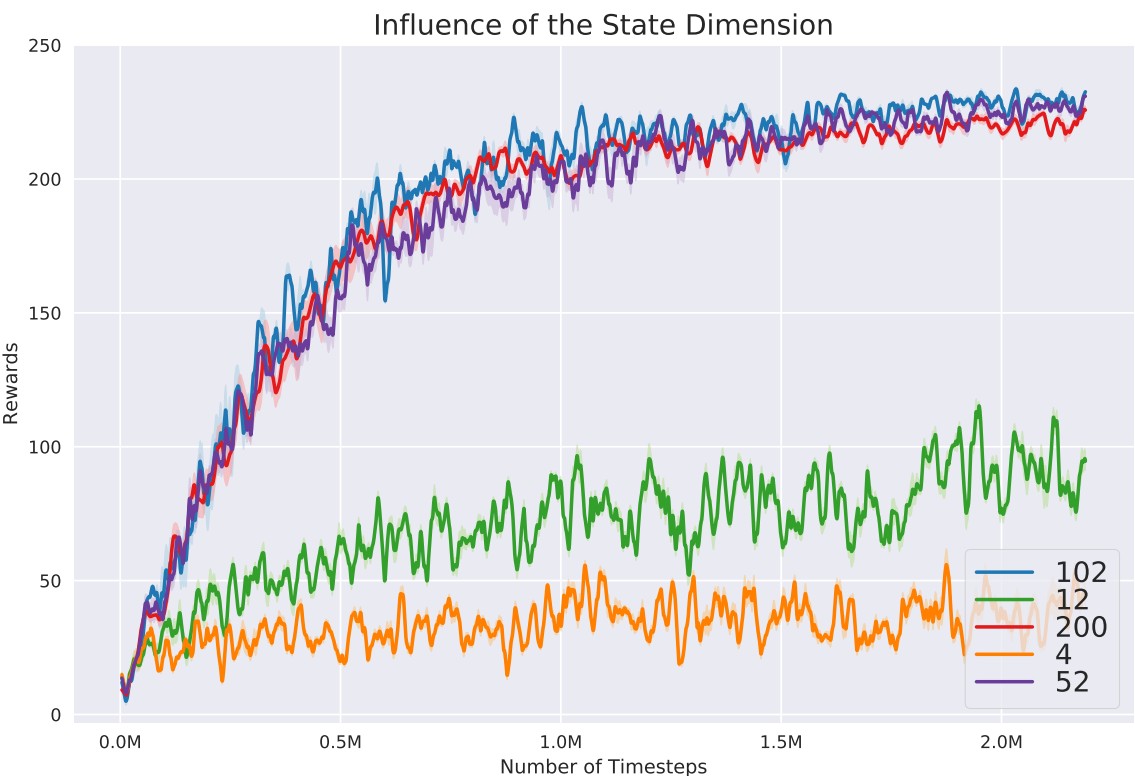

Figure 11: Influence of the state dimension (mean and standard error for 10 runs) for PPO algorithm for SRL Splits in Navigation 2D random target environment. Each label correspond to the state dimension of the model.

### B.8 INFLUENCE OF THE TRAINING SET SIZE

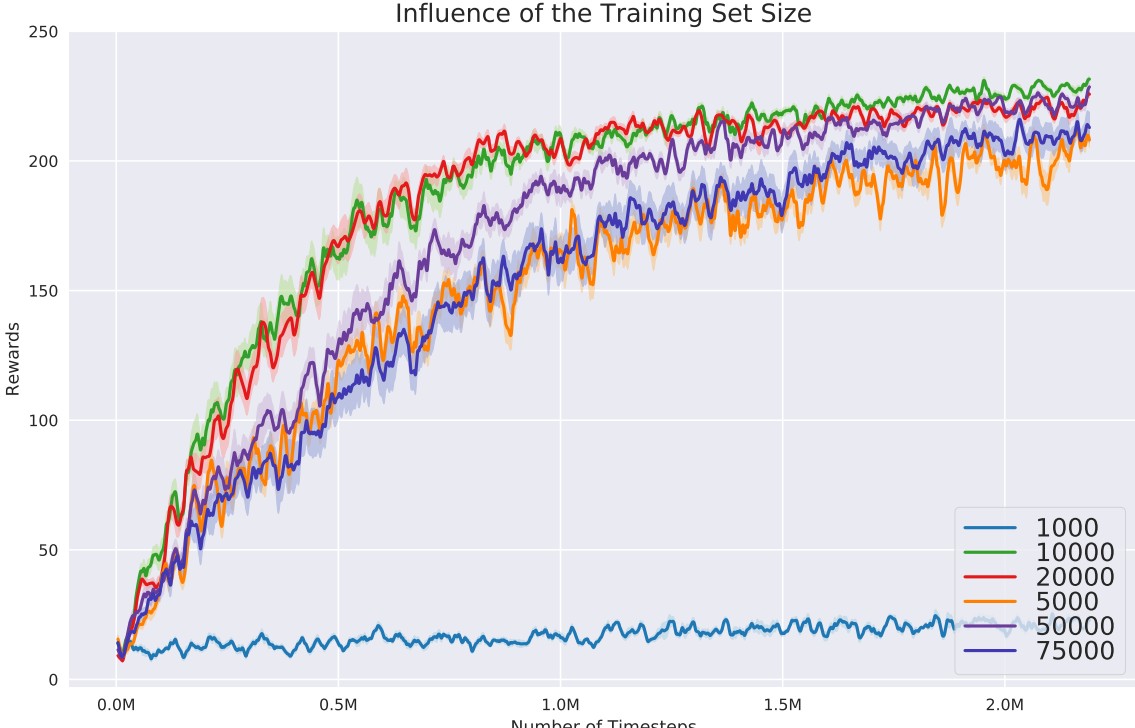

Figure 12: Influence of the training set size (mean and standard error for 10 runs) for PPO algorithm for SRL Splits in Navigation 2D random target environment. Each label corresponds to the number of samples used to train the SRL model.

The influence of the training set size (Fig. 12) is somehow similar to the influence of the state dimension (Fig. 11). A minimal number of training samples is required to solve the task, but over a certain limit, increasing the training set size is not beneficial anymore.

## C  INFLUENCE OF ROTATION ON *GTC*

In this section, we discuss the influence of rotation on the *GTC* metric in order to answer the question: how would rotating the measurement frame of the ground truth state influence the results?

First, let's start with an intuitive example of a 2D robot position. In that case, if $\alpha$ is the frame rotation angle, the new $x$ coordinate (we will write it $x'$) is $x' = x \times cos(\alpha) + y \times sin(\alpha)$. The correlation between $x$ and $x'$ should a priori decrease until $\alpha = \pi/4$ ($cos(\pi/4) = sin(\pi/4) \approx 0.71$), after which $x'$ should have a higher correlation with $y$ (and $y'$ should be more correlated with $x$). This first example uses in fact the additional assumption that $\frac{\sigma_y}{\sigma_x} \approx 1$, as we will see in the more formal answer we present in the following paragraph.

We will keep the study in $]0, \pi/2[$ because the behavior for other values of alpha can be deduced by symmetry. Under the assumption that the data are zero-centered[4] and that $x$ and $y$ are independent, we can derive these formulas for correlation between $x$, $y$ and $x'$:

$$\rho_{xx'} = \frac{cos(\alpha) \times \sigma_x^2}{\sqrt{cos^2(\alpha)(\sigma_x^2)^2 + sin^2(\alpha)\sigma_x^2\sigma_y^2}} = \frac{1}{\sqrt{1 + (tan(\alpha)\frac{\sigma_y}{\sigma_x})^2}} \tag{3}$$

$$\rho_{yx'} = \frac{sin(\alpha) \times \sigma_y^2}{\sqrt{cos^2(\alpha)(\sigma_x^2)^2 + sin^2(\alpha)\sigma_x^2\sigma_y^2}} = \frac{(\frac{\sigma_y}{\sigma_x})^2 tan(\alpha)}{\sqrt{1 + (tan(\alpha)\frac{\sigma_y}{\sigma_x})^2}} \tag{4}$$

---

[4]See https://tinyurl.com/yarqrmml

Looking at the equations, the correlation between $x$ and $x'$ ($\rho_{xx'}$) will decrease with $\alpha$, whereas $\rho_{yx'}$ will increase with it. The two coefficients are equal for $\alpha = tan^{-1}(\frac{\sigma_x^2}{\sigma_y^2})$ (equal to $\pi/4$ when $\sigma_x = \sigma_y$) with a value of $\frac{1}{\sqrt{1+(\frac{\sigma_x}{\sigma_y})^2}}$ which is around $0.71$ when $\sigma_x \approx \sigma_y$.

Hence, the two important factors are the rotation angle $\alpha$ and the ratio $\frac{\sigma_y}{\sigma_x}$. The additional assumption (that is true in our experiments) that makes GTC less sensitive to rotation is $\frac{\sigma_y}{\sigma_x} \approx 1$. If this is not the case, a simple solution is to normalize ground truth data (zero mean, unit variance).

