# OpenReview forum: "Decoupling feature extraction from policy learning: assessing benefits of state representation learning in goal based robotics"
_ICLR.cc/2019/Conference_

### Official Review · AnonReviewer2 · 2018-11-01
**Interesting experiment results, unfortunately lacking in terms of new insights.**

**Rating:** 4
**Confidence:** 4

**Review:**

The paper is easy to read and the presentation is clear, and I really appreciate this.

The authors address the very important topic of feature extraction and state representation learning. New results in this area are always valuable and welcome. However, my feeling is that the paper falls short in terms of making sufficient new contributions for an ICLR paper.

1. The authors propose to learn a state representation by either training using a combined loss function, or training several representations using multiple loss functions followed by stacking. These are standard and well-known techniques in machine learning. The key contribution one looks for is in terms of new insights on why and when each approach works. The paper fails to provide much insight in this regard. Take this simple scenario: Suppose my input image is actually generated by a linear map plus gaussian noise on the true states. Then I can simply use a PCA as my "auto encoder" and happily learn a high quality state representation close to the ground truth. We know why this works. In the real task, the image is a complex non-linear transformation of the true states. What insights do I gain from this work in terms of how I should tackle this?

2. Section 3 states some desirable characteristics in constructing a state representation. These are well-known and fundamental aspects of machine learning -- applicable to almost all models that we want to learn. In this sense, I do not find the section very informative.

3. The empirical results (say, Table 1) seem too noisy to interpret (other than that using the ground truth provides the best performance). It almost seems to suggest that one should simply use random features (as done in the "extreme learning machine" approach). Again, not much insight to draw from this.

4. Last comment. Suppose I have a new robotic goal-directed task and my inputs are camera images. Does this work tell me something that I don't already know in terms of learning new feature representation that is highly suitable for my task?

---

> ### Author Response · Authors · 2018-11-24
> **Reply to Reviewer 2**
>
>
> Dear reviewer,
> Thank you for your remarks!
>
> 1. We indeed do not have strong theoretical result on the applicability of our approach, however, we provide some insight about the way of performing efficient state representation learning in the case of goal based tasks.  In particular, we highlight the fact that auto-encoder based approaches and approaches based on action or next state prediction have complementary strengths that need to be combined to achieve good performances. The use of GTC metrics also provides better understanding of what was learned by the SRL methods and we show that this is a good proxy for the final RL performance.
>
> Although the idea of "stacking" models is not new, this is, to the best of our knowledge, the first work proposing stacking for learning a disentangled state representation.
>
>
> 2. We agree the desirable aspects are common sense. We wanted to give more context, as it is still important for us to clarify what we are looking for before proposing any solution. Following your remark, we reduced this section size to give more space for the technical description of our approach.
>
>
> 3. Table 1 should not be used alone. It only gives insights about the "sufficiency" of each method using a large budget (5 Millions steps). One should look at additional tables (with different time-steps budgets) along with the ground truth correlation (GTC, i.e., what was learned, is the representation interpretable?).
> Indeed, the evaluation of random features in comparison with other SRL approaches is one of our interesting results and we consider it as a good baseline versus end-to-end learning. Nevertheless, the first contribution of the paper is to advocate for the decoupling of policy learning from feature extraction. We hypothesize that random features should work in environments where visual observations are simple enough, and random features can preserve enough information (cf GTC tables).
> Following your remark, we updated the experiment and intro sections to clarify our results.
>
> 4. Good question. First, if you adopt reinforcement learning, this work should incite you to use a SRL model instead of learning the policy end to end. Our study will also help you choose the objective function: it is important to combine several objectives; using an inverse model or an auto-encoder alone is usually not sufficient. Finally, it gives you hints on the choice and effects of the different hyper-parameters: what state dimension is required, how many samples are needed, how to evaluate the learned states (using GTC as a proxy for RL performance) or how to choose the weights when combining objectives (and insights on the influence of changing the weights).

---

### Official Review · AnonReviewer1 · 2018-11-05
**Unclear approach and contribution**

**Rating:** 3
**Confidence:** 4

**Review:**

This paper aims at comparing end-to-end learning vs separately learning a state representation and subsequently a controller.

While this would be a relevant and important topic, the paper does not currently present consistent evidence to support this hypothesis.

In particular:
- The approach proposed in the approach is not explained in sufficient details. After reading Sec.4 I have only a very vague and high-level idea of how the proposed approach might work. In Figure.2, what is I_t? What is the model that you are training? How are you learning this model? how do you define L_inverse?
- The cited literature about state representation learning is absolutely incomplete. Papers like Lange et al. , Wahlström et al. and Finn et al. and citations herewithin.
- From the experimental results, it is difficult to say anything definitive about the proposed hypothesis. 1) There are multiple end-to-end approaches in the literature, with significant differences in performance. which one are you using? (it seem A2C and PPO, but to which label do they correspond in the tables?) 2) How do you tune the weights of the reward function proposed? This seems an important design choice, but it is not much discussed. 3) In the table reported (e.g., Table 1) it does not seem to me that SRL consistently outperforms other approaches. Even for the arm tasks, Random features seem to outperform the proposed approach (and indeed all the methods except the ground truth). What is going on there?

Overall¸ the paper would benefit from a clearer and more detailed text, and from improved experiments and comparisons.

Minor comments:
- It is unclear to me what "goal-based robotic tasks" means. How do you define a task without a goal?
- An important and missing characteristic of a suitable state representation should be the generalization. In fact, a good representation would ideally allow the agent to generalize to some degree.
- It seems very odd to me that the "action should be implicitly encoded into the state representation" could you elaborate of the motivation for this and the effects?

References:
- Autonomous reinforcement learning on raw visual input data in a real-world application
S Lange, M Riedmiller, A Voigtlander
Neural Networks (IJCNN), The 2012 International Joint Conference on, 1-8
- From pixels to torques: Policy learning with deep dynamical models
N Wahlström, TB Schön, MP Deisenroth
arXiv preprint arXiv:1502.02251
-  Deep Visual Foresight for Planning Robot Motion
Chelsea Finn, Sergey Levine
International Conference on Robotics and Automation (ICRA), 2017

---

> ### Author Response · Authors · 2018-11-24
> **Reply to Reviewer 1**
>
>
> Thank you for pointing out the missing details about the approach, we have updated Sec 4.4:
> - $I_t$ refers to the input image, that will be used in the reconstruction loss.
> - In our experiments, $L_{inverse}$ is defined as a cross entropy loss. As we are in a discrete action setting, we treat the problem of predicting the taken action as a classification problem. For continuous actions, we would treat that problem as a regression.
> Concerning the model (architecture, how is the training done), implementation details can be found in Appendix A.
>
> We updated the related work with Lange et al. and Wahlström et al. and pointed to Lesort et al. 2018 for a more complete overview of SRL. Concerning Finn et al. we decided to not add it, because even if the paper is about robotics policy learning, learning a representation is not presented as a major component of the approach.
>
>
> Concerning the experimental results:
>
> 1) As written in the different figures/tables captions and as stated in Appendix A: ``PPO was the RL algorithm that worked well across environments and methods without any hyper-parameter tuning, and therefore, the selected one for our experiments''. We used PPO in all experiments except in an additional one (Fig.~7) where we wanted to show the effect on the performance of choosing another RL algorithm.
>
> 2) Indeed, choosing the right weights seems to be crucial. As stated in Sec 4.3 ``Because we consider each objective to be relevant, we chose the weights such that they provide gradients with similar magnitudes.'' and in Sec 4.4 ``To have the same magnitude for each loss, we set $w_{reconstruction}=1$, $w_{reward}=1$ and $w_{inverse}=2$.''. In the appendix B.6, we dedicated a whole experiment to explore the use of different weights and show that the method is quite robust to weight changes as long as they are in the same order of magnitude. Following your remark this experiment is now part of the main experiment section.
>
>
> 3) Table 1 only shows the final performance of all methods at the end of training using a large budget (5 Millions steps). It gives an insight of the sufficiency of each method. One should not focus only on that table but also look at additional metrics such as the GTC and performance for a fixed budget. We recognize that this aspect was not clearly presented, and we updated the experiment section to reflect the usefulness of SRL in terms of sample efficiency (we give the performance for different fractions of the total budget).
>
> The fact that random features are a good baseline is also part of our results; we don't postulate that our proposed method outperforms all other methods, but rather advocate for using SRL instead of end-to-end learning.
> For the robotic arm, GTC (cf Table 8) suggests that random features captured as much information of the ground truth states as the other methods. We also already provide possible reasons in the experiment section for that.
>
> We updated the paper to clarify what we mean by "goal-based robotic tasks":
> 1. The controlled agent is a robot
> 2. The reward is sparse and only depends on the previous state and taken action, not on a succession of states
> Overall, our definition includes tasks presented in the HER paper (https://gym.openai.com/envs/#robotics) and excludes settings such as "running" or "execute a back flip".
>
>
> About the generalization of state representation: We agree that it is an important characteristics. In the current paper we only tackle  the 'interpolation' aspect of generalization, however generalization to new environments/tasks/robots would be an interesting future work. We added a discussion about it in the paper.
>
>
> "action should be implicitly encoded into the state representation"
>
> As explained in the paper, using a reward prediction objective (giving current state, action and predicting the reward) alone yields a "state representation with one cluster per reward value". That is to say, because this is a classification objective, in the case of sparse reward, the solution found to minimize the loss will be to separate the data into n groups (where $n=3$ in our case, because we have either positive, negative or null rewards). The problem with that type of representation is that it does not have any structure that the agent can benefit from (e.g., nothing enforce that neighbours in the ground truth state space are  neighbours in the learned states). To mitigate that potential issue, and because we are in the context of MDPs ($s_{t+1} = f(s_t, a_t)$), we suggest to predict the reward $r_t$ from $s_t$ and $s_{t+1}$ instead of $s_t$ and $a_t$. The idea is that if the reward depends on the action $a_t$, because we only give $s_t$ and $s_{t+1}$, it should also encode $a_t$ in the state representation to be able to minimize the loss, therefore enforcing some structure.

---

### Official Review · AnonReviewer3 · 2018-11-07
**This paper on SRL provides some interesting results, but it methods should be better motivated, and its conclusions be made more precise..**

**Rating:** 5
**Confidence:** 4

**Review:**

This paper discusses State Representation Learning for RL from camera images. Specifically, it proposes to use a state representation consisting of 2 (or 3) parts that are trained separately on different aspects of the relevant state: reward prediction, image reconstruction and (inverse) model learning. The paper is easy to read, and seems technically sound. However, the conclusions do not directly follow from the results, so should be made more precise. The contribution is minor, and the reasoning behind it could be better motivated.

The most important point of critique is that the conclusion that the split representation is the best is at best premature. The presented results indicate that SRL is useful (Table 1), and that auto-encoding alone is often not enough. Other than that, the different approaches tested all work well in different tasks. The discussion of the results reflects this, but the introduction and conclusion suggest otherwise.

The same problem also occurs for the conclusion about the robustness of SRL approaches. In the main text, no results are presented that warrant such a conclusion. The appendix includes some tests in this direction, but conclusions should not be based on material that is only available in the appendix. Furthermore, even the tests in the appendix are not comprehensive enough to to warrant the conclusion as written.

The second point is the motivation of the split approach: it seems in direct contradiction with the "disentangled" and "compact" demands the authors pose. Because the parts of the state that are needed for multiple different prediction tasks (reconstruction, inverse model, etc.) need to be in the final
state representation multiple times. Due to the shared feature extractor, the contradictory objectives (and hence the need for tuning of the weights in the cost function) are still a potential problem.

Minor points:

- The choice for these tasks is not motivated well. Please indicate why these tasks are chosen. It seems the robot arm task is very similar to the navigation task, due to robot arm's end effector being position controlled directly. Why is it worthwhile to study this task separately?

- The GTC metric is not very well established (yet). Please provide some extra information on how it is calculated. This should also include some discussion on why this metric allows judging sufficiency and disentangledness. How would rotating the measurement frame of the ground-truth influence the results?

- Why are the robotics priors not in Table 1?

---

> ### Author Response · Authors · 2018-11-24
> **Reply to reviewer 3**
>
> Thank you for your very interesting remarks.
>
> "Other than that, the different approaches tested all work well in different tasks."
>
> That is true that in our case, SRL does not necessarily improve the final results; however, it is useful to improve the learning speed of the policy. We updated the experiment section to be clearer.
>
> "conclusions should not be based on material that is only available in the appendix"
> Following your remark, we revamped the experiment section and updated introduction and conclusion.
>
> "the motivation of the split approach seems in direct contradiction with the "disentangled" and "compact" demands the authors pose"
>
> Very good point. The "compact" requirements still holds, as the SRL model has a much smaller dimension compared to the raw sensor data. We obtained compact and disentangled features for robot position using an inverse dynamics model. However, for the goal position encoding in the presented environments, a smaller dimension can be achieved in theory, so we agree that work remains to do in that direction.
>
> The second problem that you expose is the redundancy of information. This is true that in our current setting, some information has to be encoded twice (e.g., the robot position by the reconstruction and the inverse dynamics losses). However, the Split model still favors to untangle factors of variation because of the parts that minimize different objectives. Encoding an information twice does not hurt the performance as long as the factors of variation are untangled.
> Following your remark, we updated our definition of "disentanglement" to reflect that this second aspect is more related in our opinion to compactness.
>
> While this is not reported in the paper, during our experiments, we tested a variant of the Split model to remove that redundancy: instead of masking states that are not optimized by an objective, we give the full state to each objective but only use the fraction of the gradients that correspond to each split. For example, with that variant, the decoder (of the reconstruction loss) has access to the state encoded by the inverse dynamics loss, that corresponds to the controlled agent, but cannot optimize it. As a result, the encoder of the reconstruction loss does not have to encode the controlled agent a second time. We did not keep this variant because this lead to instabilities in the optimization process (possible reason: the gradients that are not back-propagated are still computed and can have a greater magnitude that ones that are back-propagated).
>
> We agree that the SRL Split approach only mitigates the conflicting objectives problem. Even though the feature extractor is shared, "splits" can use separate parts of the network to minimize their objectives. To remove completely the conflicting objectives issue, a solution is to train different feature extractors for each part of the state representation. However, this requires extra memory (n times more if n is the number of splits) and does not scale with the number of splits. Therefore, the proposed approach can be seen as a compromise between mixing objectives and training separate models.
>
> " The choice for these tasks is not motivated well."
>
> We updated the environment section to motivate our choice.
> The task with the arm is actually similar to the 2D navigation, but remains interesting to tackle separately. First, we consider it to be harder to solve, as the input is only a 2D image for a 3D task and the minimal number of dimension to encode the information is higher. The visual saliency is also quite different compared to the navigation task: here the arm and the target are much more salient than in the mobile robot task. This allows to study the behavior of method like auto-encoder that rely only on the image (compared to method like inverse model that uses additional information).
>
> Concerning the GTC metric, we updated the paper with the full formula and added a discussion on that metric.
>
> We also added a section in the appendix to answer your interesting question on how does rotation affects GTC metric. The takeaway is that as long as the ground truth coordinates distributions have similar variance, the correlation will remain high. If the condition does not hold, normalizing the ground truth data (zero mean, unit variance) should solve this issue. Finally, one should note that the condition is satisfied in our experiments.
>
> During our preliminary experiments, robotic priors showed bad performances inherent to the way they work: despite ignoring moving distractors, in presence of a static object initialized at different positions for each episode, this method will create in the state representation a cluster per episode. This prevent generalization and good performances in the presented RL tasks. Because of that, we chose to focus on the other methods that did not suffer from this issue and did not run exhaustive experiments on all the tasks, and therefore present only partial results.

---

### Author Response · Authors · 2018-11-24
**Paper Update Following Reviewers Remarks**

Thanks to the reviewers interesting remarks, we updated the paper. We made the following changes:

- we updated introduction/conclusion to clarify our contributions
- we added a discussion on the GTC metric and on the choice of the environments
- we updated the experiment section to clarify our results (some results were moved from appendix to main section)
- we removed the section on "Aspects of an adequate method"
- we updated the requirements section to include "generalization"
- we updated the related work to include missing references and pointed to a recent survey on SRL
- we added details to the "SRL Split" section in order to clarify how were computed the losses
- we added a section in the appendix to discuss the influence of rotation on the GTC metric

---

### Meta-Review · Area_Chair1 · 2018-12-14

**Confidence:** 4
**Recommendation:** Reject

**Metareview:**

This paper proposes a new method for combining previous state representation learning methods and compares to end-to-end learning without without separately learning a state representation. The topic is important, and the authors have made an extensive effort to address the reviewer's concerns, particularly regarding clarity, related work, and accuracy of the drawn conclusions. The reviewers found that the main weakness of the paper was the experiments not being sufficiently convincing that the proposed approach is better than the alternatives. Hence, it does not currently meet the bar for publication.